# Prognostic Value of Telomeric Zinc Finger-Associated Protein Expression in Adenocarcinoma and Squamous Cell Carcinoma of Lung

**DOI:** 10.3390/medicina57111223

**Published:** 2021-11-10

**Authors:** Gun-Jik Kim, Jae-Ho Lee, Mincheol Chae, Deok-Heon Lee

**Affiliations:** 1Department of Thoracic and Cardiovascular Surgery, School of Medicine, Kyungpook National University, 130 Dongdeok-ro, Jung-gu, Daegu 41944, Korea; straightroot@knu.ac.kr; 2Department of Anatomy, Keimyung University School of Medicine, Dageu 42601, Korea; anato82@dsmc.or.kr; 3Dongsan Medical Center, Department of Thoracic and Cardiovascular Surgery, Keimyung University School of Medicine, Dageu 42601, Korea; ysacmc@naver.com

**Keywords:** TZAP, telomere, lung cancer, ZBTB48

## Abstract

*Background and Objectives*: Telomeric zinc finger-associated protein (TZAP) is a telomere regulation protein, previously known as ZBTB48. It binds preferentially to elongated telomeres, competing with telomeric repeat factors 1 and 2. TZAP expression may be associated with carcinogenesis, however; this study has not yet been performed in lung cancer. In this study, we examined the clinicopathological and prognostic values of TZAP expression in non-small cell lung cancer (NSCLC). *Materials and Methods:* Data were collected from The Cancer Genome Atlas. The clinical and prognostic values of TZAP for NSCLC were examined in adenocarcinoma (AD) and squamous cell carcinoma (SCC). *Results:* TZAP expression significantly increased in NSCLC tissues compared with normal tissues. In AD, TZAP expression was lower in patients with higher T stage (*p* = 0.005), and was associated with lymph node stage in SCC (*p* = 0.005). Survival analysis showed shorter disease-free survival in AD patients with lower TZAP expression (*p* = 0.047). TZAP expression did not have other clinical or prognostic value for AD and SCC. *Conclusions*: TZAP expression is a potential prognostic marker for NSCLC, especially in patients with AD.

## 1. Introduction

Telomeres, composed of TTAGGG repeated sequences, are nucleoprotein complexes that cap the ends of eukaryotic chromosomes [1]. Telomeres in normal somatic cells are shortened by approximately 0–200 base pairs during cell division. The critical length initiates replicative senescence or apoptosis [2,3]. Therefore, telomere regulation within an optimal length is necessary for cellular process [1,2,3]. In cancer cells, telomere shortening is inhibited by telomerase reverse transcriptase (TERT), whereas the remaining cells maintain telomere length using alternative lengthening mechanisms [4,5]. As a result of this process, longer telomeres are created and are cut back to previous levels by a shortening process [6]. A current study showed that ZBTB48, as a zinc finger protein, now known as telomeric zinc finger-associated protein (TZAP), rapidly shortens telomeres [7]. Although this suggests that TZAP regulation may influence cancer pathogenesis, it has not been evaluated in various cancers.

Lung cancer is the leading risk factors of cancer-related deaths, and has poor survival rates [8]. Many causes of lung cancer are well-known, such as smoking, occupational agents, radiation, and environmental pollutants [9,10]. They influence telomere regulation in human diseases, especially in non-small cell lung cancer (NSCLC). It was controversial that deregulation of telomere length has a prognostic value for patients with NSCLC, and may therefore aid clinicians in making therapeutic decisions [11,12]. Therefore, we analysed TZAP expression in NSCLC for the first time.

Recently, advanced genomic profiling using next-generation sequencing allowed to recognize the genetic characteristics of cancer. Large-scale cancer genome studies, such as The Cancer Genome Atlas (TCGA), have been used to investigate genes in different cancer types [13,14]. More so, TCGA can investigate specific histological types of lung cancer, as adenocarcinoma (AD) and squamous cell carcinoma (SCC), using histological data and clinical parameters. Therefore, our primary goal was to study the clinicopathological and prognostic value of TZAP expression in NSCLC using RNA-seq gene expression data obtained from TCGA datasets.

## 2. Materials and Methods

Primary data from TCGA data portal were downloaded in March 2021. The TCGA dataset consisted of 1130 samples, including 1019 primary tumor tissues (517 AD and 502 SCC) and 111 normal solid tissues taken adjacent to the tumor. During the RNA-seq data analysis of NSCLC, AD and SCC datasets were sorted from TCGA with TZAP mRNA expression and clinical parameters. This study followed the publication guidelines for using TCGA datasets (http://www.cancer.gov/about-nci/organization/ccg/research/structrual-genomics/tcga/using-tcga/citing-tcga, accessed on 8 March 2021). Overall survival (OS) and disease-free survival (DFS) was the duration from the date of biopsy to the date of the last follow-up visit or to the date of death due to any cause and to the date of any type of recurrence, respectively.

For statistical analyses, the Statistical Package for the Social Sciences (SPSS), version 24.0, for Windows (IBM, Armonk, NY, USA) was used. Chi-square and Mann–Whitney U tests were performed for categorical and continuous variables, respectively. For survival analysis, the mean gene expression was used as a cut-off to divide the patients into high- and low-expression groups. Survival analysis was performed using the Kaplan–Meier method, and the log-rank test was used to identify statistically significant differences between the two groups. Statistical significance was defined as a two-tailed *p*-value < 0.05.

## 3. Results

TZAP expression was statistically higher in NSCLC than in normal tissues (Figure 1). In AD, the median level of TZAP expression was 8.65 ± 0.59, which was statistically higher than that in normal tissues (8.35 ± 0.32, *p* < 0.001). In SCC, TZAP expression was significantly different (8.56 ± 0.54 vs. 8.40 ± 0.38; *p* = 0.027).

The clinical characteristics according to TZAP mRNA expression in AD and SCC are summarized in Table 1 and Table 2, respectively. TZAP expression was lower in AD patients with higher T stage (*p* = 0.005), higher N stage (*p* = 0.065), and epidermal growth factor receptor (EGFR) mutation positivity (*p* = 0.066), although the N stage and EGFR mutation were not statistically significant. This indicated that lower TZAP expression had an association with clinical characteristics, inducing poorer prognosis.

In SCC, lower TZAP expression was observed in patients with a higher N stage (*p* = 0.005). It was also associated with cancer metastasis (51.4% vs. 85.7%); however; it did not have significance (*p* = 0.124). TZAP expression was not associated with other characteristics in patients with AD.

For survival analysis, follow-up was performed during 2622 ± 245 days (range: 4–7062 days) in AD and 2128 ± 126 days (range: 2–5287 days) in SCC, respectively. Univariate survival analysis revealed that OS in AD patients was not associated with TZAP expression (2595.66 ± 336.70 vs. 2679.27 ± 332.14 days, *p* = 0.131; Figure 2A); however, a shorter DFS was found in AD patients with lower TZAP expression (2422.98 ± 490.78 vs. 2833.90 ± 372.97 days, *p* = 0.047; Figure 2B).

In SCC, TZAP expression did not have any prognostic values of OS (2215.91 ± 174.72 vs. 1946.47 ± 148.77 days, *p* = 0.805; Figure 3A) and DFS (2614.32 ± 215.95 vs. 2812.18 ± 218.12 days, *p* = 0.679; Figure 3B).

## 4. Discussion

This is the first study to examine TZAP mRNA expression in NSCLC. TZAP expression may be a chief factor for telomere regulation in cancers [7]. When its expression is not sufficient in cancer cells, cells with longer telomeres can easily change into immortal cells, anticipating cancer development [4,5]. In this way, TZAP expression could mediate telomere trimming, thereby restricting abnormally long telomeres [15,16,17,18,19,20]. However, TZAP expression has not been documented in various cancers, thus, this hypothesis should be confirmed.

The Cancer Genome Atlas datasets showed that TZAP mRNA expression may play a significant role in pancreatic and colorectal cancers [17]. Here, we recognized the clinical and prognostic value of TZAP mRNA expression in NSCLC. Previous studies have suggested that telomere regulation plays a major role in NSCLC pathogenesis [11,12]. As expected, we found that TZAP expression was lower in AD patients with higher T and N stages, suggesting poorer prognosis. Moreover, it tended to be associated with EGFR mutations. Our previous study showed that telomere length was longer in AD than in SCC (45% vs. 22%), although the difference did not get statistical significance (*p* = 0.100) [12]. These results were consistent with the hypothesis that insufficient TZAP expression could lead to the generation of long telomeres in AD, thus promoting cancer development via immortal cells [7,12,18]. In SCC, lower TZAP expression is related to lymph node invasion; however, it does not have other clinical characteristics and prognostic values. Its precise mechanism is still unclear and should be examined in other cancers.

Insufficient TZAP expression may induce longer telomere length [20]. A strong correlation between longer telomeres and increased risk of NSCLC, especially AD, has been reported [21,22,23]. Moreover, telomerase activity and telomere regulation gene polymorphisms are associated with a higher risk of adenocarcinoma and longer telomere length, suggesting that patients with adenocarcinoma may be affected by telomerase activity and telomere length [11,12,21,22,23]. Therefore, genetic studies of telomere regulation genes, such as TERT, telomeric repeat-binding factor 2 (TRF2), and TZAP, should be added to explain the mechanism of AD pathogenesis. Additionally, the patients enrolled in this study received different therapeutics after the surgery; therefore, this difference may influence the prognosis regardless of TZAP expression.

It is the limitation of this study using bigdata, and further studies should clarify mechanisms of telomere regulation via TZAP expression in cancer patients.

## 5. Conclusions

We studied the expression of TZAP in patients with NSCLC. Telomeric zinc finger-associated protein changes have great significance as clinical and prognostic markers in AD. The present study warrants future molecular study to clarify the mechanisms of TZAP for its clinical potential.

## Figures and Tables

**Figure 1 medicina-57-01223-f001:**
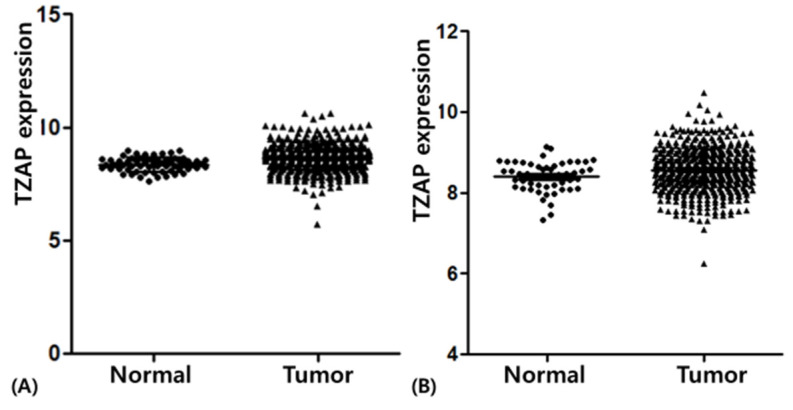
TZAP expression in lung cancer and normal tissues: (**A**) adenocarcinoma; (**B**) squamous cell carcinoma.

**Figure 2 medicina-57-01223-f002:**
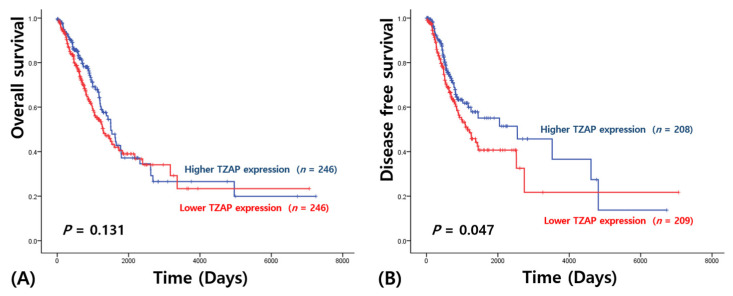
Survival analysis in adenocarcinoma: (**A**) overall survival; (**B**) disease-free survival.

**Figure 3 medicina-57-01223-f003:**
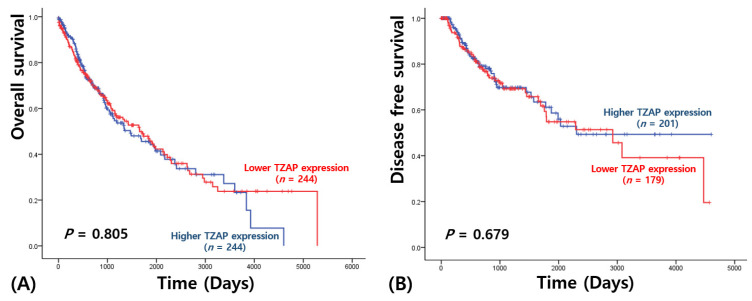
Survival analysis in squamous cell carcinoma: (**A**) overall survival; (**B**) disease-free survival.

**Table 1 medicina-57-01223-t001:** Clinical characteristics of TZAP mRNA expression in lung adenocarcinoma.

	TZAP Expression		*p* Value
	High (%, *N*)	Low (%, *N*)	
Age			0.558
<65 years	51.3 (121)	48.7 (115)	
≥65 years	48.6 (124)	51.4 (131)	
Sex			0.320
Male	47.6 (107)	52.4 (118)	
Female	52.1 (139)	47.9 (128)	
T stage			0.005
T1	57.2 (95)	42.8 (71)	
T2	50.0 (130)	50.0 (130)	
T3	35.6 (16)	64.4 (29)	
T4	22.2 (4)	77.8 (14)	
N stage			0.065
N0	53.1 (169)	46.9 (149)	
≥N1	44.2 (72)	55.8 (91)	
M stage			0.579
M0	47.5 (154)	52.5 (170)	
M1	41.7 (10)	58.3 (14)	
EGFR mutation			0.066
(+)	48.3 (14)	51.7 (15)	
(-)	31.0 (57)	69.0 (127)	

TZAP: telomeric zinc finger-associated protein; EGFR: epidermal growth factor receptor.

**Table 2 medicina-57-01223-t002:** Clinical characteristics of TZAP expression in lung squamous cell carcinoma.

	TZAP Expression		*p* Value
	High (%, *N*)	Low (%, *N*)	
Age			0.114
<65 years	45.5 (86)	54.5 (103)	
≥65 years	52.8 (158)	47.2 (141)	
Sex			0.215
Male	48.3 (175)	51.7 (187)	
Female	54.8 (69)	45.2 (57)	
T stage			0.399
T1	53.2 (58)	46.8 (51)	
T2	47.6 (136)	52.4 (150)	
T3	57.1 (40)	42.9 (30)	
T4	43.5 (10)	56.5 (13)	
N stage			0.005
N0	55.0 (171)	45.0 (140)	
≥N1	41.5 (71)	58.5 (100)	
M stage			0.124
M0	48.6 (195)	51.4 (206)	
M1	14.3 (1)	85.7 (6)	

## Data Availability

The datasets used and/or analyzed during the current study are available from the corresponding author.

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
