# Peer review of "Prognostic Value of Telomeric Zinc Finger-Associated Protein Expression in Adenocarcinoma and Squamous Cell Carcinoma of Lung"

_medicina, 2021, doi:10.3390/medicina57111223_

Round 1
Reviewer 1 Report
Line 49; reference 8 - I would use more recent data for cancer statistics. Please refer to this link https://pubmed.ncbi.nlm.nih.gov/33433946/.
Line 49 - I would replace 'causes' with 'risk factors'.
Lines 52-54; reference 11: I have to disagree with the statement. As per the abstract of article referenced number 11 "Survival analysis showed no prognostic value for rs401681 polymorphisms or telomere length in NSCLC." But this holds true for reference 12. I would suggest correcting the statement or using that reference elsewhere.
Line 72: Do you mean date of diagnosis through biopsy when you say 'date of surgery'. If that's the case. I would correct the statement to reflect the same. The same goes with disease-free survival.
Lines 98-99: How can you be certain about the statement. Your data shows association but I am not sure if you can extrapolate that to causation (I would using the term 'inducing').
Line 146: I would recommend changing 'inducing' to 'suggesting'.
Author Response
Letter to referees
First of all we would like to thank the referees and editor for their time in reviewing our manuscript. Their constructive comments helped us to improve our work. We submit the revised version for your consideration. Please reconsider below a detailed response to your comments.
-----------------------------------------------------------------------------
Line 49; reference 8 - I would use more recent data for cancer statistics. Please refer to this link https://pubmed.ncbi.nlm.nih.gov/33433946/.
-->It is better. This reference was changed.
Line 49 - I would replace 'causes' with 'risk factors'.
-->It is better. it was changed.
Lines 52-54; reference 11: I have to disagree with the statement. As per the abstract of article referenced number 11 "Survival analysis showed no prognostic value for rs401681 polymorphisms or telomere length in NSCLC." But this holds true for reference 12. I would suggest correcting the statement or using that reference elsewhere.
--> These two results were controversial, therefore, this description was added.
Line 72: Do you mean date of diagnosis through biopsy when you say 'date of surgery'. If that's the case. I would correct the statement to reflect the same. The same goes with disease-free survival.
--> The date of surgery and biopsy was same day. It was corrected.
Lines 98-99: How can you be certain about the statement. Your data shows association but I am not sure if you can extrapolate that to causation (I would using the term 'inducing').
--> I am agree with our comment, and it was corrected.
Line 146: I would recommend changing 'inducing' to 'suggesting'.
--> I am agree with our comment, and it was corrected.
Thank you for your constructive and favorable reviews.
Sincerely yours,
Reviewer 2 Report
In this article authors analyze the relation between telomeric zinc-finger associated protein (TZAP) expression and outcomes in patients with lung adenocarcinoma and squamous cell carcinoma. The correlation between the expression of corresponding protein and development of different cancers, their malignancy and clinical outcome has been studied and published before. Dos Santos et al., Mol Biol Res Commun. 2021 have also studied this relationship in lung adenocarcinoma. However this analysis has not been done with squamous cell carcinoma of lungs which is the novelty of this manuscript. The level of TZAP expression can be linked to a poor prognosis in some type of cancers, but in other type of cancers this can lead to a better prognosis. In this study authors have shown that in adenocarcinoma, TZAP expression was lower in patients with higher T stage, and was associated with lymph node stage in squamous cell carcinoma. In addition, adenocarcinoma patients with lower TZAP expression had shorter disease-free survival.
Quesitons:
- From a clinical point of view it would be interesting to know how TZAP expression affects drug therapy effectiveness. Did all included patients receive the same therapy? Maybe the TZAP expression affects the effectiveness of denoted anticancer therapy. If it is possible I would suggest dividing patients according to therapy. I think that would be a great novelty of this manuscript if some patients with altered TZAP expression would respond differently to some anticancer therapy.
- Why did authors decided to include also patients which died due to any cause in survival analysis ? I think it is essential how many patients died of cancer not other diseases to make a conclusions about TZAP expression and survival. How many such patients (died due to any cause) have been included in the study? This is important as the p value for survival analysis is very close to 0.05.
- Was there any normality test performed before the Mann–Whitney test was used?
Author Response
Letter to referees
First of all we would like to thank the referees and editor for their time in reviewing our manuscript. Their constructive comments helped us to improve our work. We submit the revised version for your consideration. Please reconsider below a detailed response to your comments.
--------------------------------------------------------
- From a clinical point of view it would be interesting to know how TZAP expression affects drug therapy effectiveness. Did all included patients receive the same therapy? Maybe the TZAP expression affects the effectiveness of denoted anticancer therapy. If it is possible I would suggest dividing patients according to therapy. I think that would be a great novelty of this manuscript if some patients with altered TZAP expression would respond differently to some anticancer therapy.
--> Patients in Big data received different anticancer therapy and detail data was not enough. Therefore, exact effect of TZAP expression can not be clear in this point. It is the limitation of this study, and its description was added in Discussion.
- Why did authors decided to include also patients which died due to any cause in survival analysis ? I think it is essential how many patients died of cancer not other diseases to make a conclusions about TZAP expression and survival. How many such patients (died due to any cause) have been included in the study? This is important as the p value for survival analysis is very close to 0.05.
--> The prognosis of cancer patients is judged by survival result of overall survival day and recur days. Therefore, both overall survival and disease free survival were analyzed.
- Was there any normality test performed before the Mann–Whitney test was used?
--> To analyze the difference of median value of TZAP expression, its distribution was checked at first. The number of normal tissue was not enough compared to tumor samples, and Mann–Whitney test as nonparametric method was performed.
Thank you for your constructive and favorable reviews.
Sincerely yours,